# The Cytotoxic Effects of Human Mesenchymal Stem Cells Induced by Uranium

**DOI:** 10.3390/biology13070525

**Published:** 2024-07-16

**Authors:** Yi Quan, Xiaofang Yu

**Affiliations:** 1Institute of Nuclear Physics and Chemistry, China Academy of Engineering Physics, Mianyang 621900, China; yuxf@caep.cn; 2Collaborative Innovation Center of Radiation Medicine of Jiangsu Higher Education Institutions, Suzhou 215000, China

**Keywords:** uranium cytotoxicity, stem cell, gap junctional intercellular communication

## Abstract

**Simple Summary:**

Uranium, a fuel material widely used for nuclear reaction and a source of nuclear weapons, is an emerging pollutant threatening human health. This study investigates the impact of uranium poisoning on mesenchymal stem cells (MSCs), which are crucial for bone formation and healing. The role of gap junctional intercellular communication (GJIC) in uranium toxicity was also examined. The study found that increasing concentrations of uranyl nitrate led to greater damage to MSCs, as shown by TEM images and confirmed by cellular viability and DSB production. Apoptosis was notably higher at a concentration of 84 μM, causing significant membrane disruption. Interestingly, closely connected cell groups were more resistant to uranium toxicity compared to sparsely grown ones, which were weakened by the use of a GJIC inhibitor. Subsequently, an impairment of wound healing and connexin expression were observed after exposure to uranyl nitrate, which was more remarkable with the increase in concentration. This suggests that GJIC, which is linked to the integrity of cellular membranes, plays a role in the response to uranium toxicity.

**Abstract:**

Bone is a major tissue for uranium deposition in human body. Considering mesenchymal stem cells (MSCs) play a vital role in bone formation and injury recovery, studying the mechanism of MSCs responding to uranium poisoning can benefit the understanding of bone damage and repair after uranium exposure. Cellular structural alterations were analyzed via transmission electron microscopy (TEM). Changes in cellular behaviors were assessed through cellular viability, apoptosis, and the production of DNA double-strand breaks (DSBs). In addition, the influence of gap junctional intercellular communication (GJIC) on uranium toxicity was assessed. The disruption of MSCs was elevated with the increase in uranyl nitrate concentration, as shown by TEM micrograph. This was verified by the results of cellular viability and DSB production. Interestingly, the results of apoptosis assay indicated significant apoptosis occurred, which was accompanied with an obvious disruption of cellular membranes. Furthermore, closely contacted cell confluence groups exhibited resistant to uranium poisoning in contrast to sparse growth groups, which can be eliminated with the pretreatment of a GJIC inhibitor in the close connection group. To verify the association between GJIC and cytotoxic effects of uranyl nitrate, GJIC function was evaluated by wound healing and cellular migration. The results showed an inhibition of the healing ratio and migration ability induced by the exposure of uranyl nitrate. The low transfer efficiency of the dye coupling experiment and depressed expression of gap functional protein connexins confirmed the impairment of GJIC function. These results suggest that uranium toxicity is involved with GJIC dysfunction.

## 1. Introduction

With the prompt development of nuclear energy technology, worldwide concerns are being raised about the environmental contamination of radioactive isotopes due to their hazardous effects on organisms. Uranium, a fuel material widely used for nuclear reaction and a source of nuclear weapons, is an emerging pollutant threatening human health [1]. Once absorbed by the human body, uranium exists in liquid as uranyl salts or bonds with biological proteins to accumulate in different organs including bones, kidneys, lungs, liver, and spleen [2,3,4]. It has been demonstrated that bone is a main tissue for uranium accumulation [5]. In addition, human osteoblasts are sensitive to uranium exposure because modifying osteoblast phenotypes, decreasing alkaline phosphatase activity and enhancing reactive oxygen species (ROS) and genomic instability, can be evidently induced by uranium [6,7,8]. Impairment of bone growth and formation resulting from inhibition of endochondral ossification was also found in mice after exposure to uranium [9,10,11]. When bone injury occurs, mesenchymal stem cells (MSCs) can be activated to repair this damage as MSCs have the potential to differentiate into adipocytes, chondrocytes, and osteoblasts instead of these damaged cells [12]. Furthermore, stem cells with a long life span probably accumulate abundant mutations to drive carcinogenesis [13,14]. Recently, extensive research has been carried out to clarify the relationship between stem cells and carcinogenesis induced by environmental toxic factors, such as ionizing radiation [15,16]. Therefore, the study of MSCs responding to uranium exposure can benefit the understanding of the physiological process involved in bone tissue damage induced by uranium.

Signal transduction between subcellular organelles becomes a vital factor to determine the outcome of cytotoxicology. For example, in the studies of non-targeted effects induced by radiation, it was found that communication among cells through physics connections or signal molecules released by irradiated cells in medium is significant for regulating the occurrence of radiation effects in non-targeted cells [17]. Reactive oxygen species (ROS) is an important signal molecule that regulates cellular physiological processes. Studies of uranium toxicity have also indicated that ROS are related to the imbalance of redox homeostasis, are associated with mitochondrial dysfunctions, and have an impact on uranium poisoning [18,19]. These radicals can threaten DNA and result in the formation of DNA single- or double-strand breaks (SSB or DSB) [20,21]. Furthermore, uranium can interact with some crucial proteins and enzymes that impact oxidative stress, cytoskeleton structure, reproduction, and energy metabolism [22,23,24,25]. For example, uranium influences osteoblast cell metabolism via inhibiting the activity of alkaline phosphatase [7]. Chronic intake of uranium in drinking water can significantly modify the bone-related gene profiles by influencing the reception of vitamin D [26]. Vitamin D_3_, β-glycerophosphate, and ascorbic acid were also reported to mediate MSCs differently from osteoblasts [27]. It also has been reported that nanoparticles increased connexin43-mediated gap junctional intercellular communication (GJIC) through activation of ROS in HaCaT cells [28]. Gap junctions have been demonstrated to constitute a family of specialized proteins called connexins (Cxs), and GJIC provides a way for communication through the transfer of small molecules (e.g., ions, signaling molecules, and secondary messengers) between neighboring cells. It stimulates many signaling pathways to regulate cell behaviors (e.g., apoptosis, proliferation, and differentiation) [29,30,31]. Considering the contribution of MSCs to differentiate into mesodermal derivations (osteoblasts, adipocytes, and chondrocytes) and regenerate heterotopic bone tissue depending on signaling transduction, it is significant to clarify the influence of GJIC function induced by uranium in MSCs. Furthermore, the mechanism of MSC damage induced by uranium is unclear and its impact on the repair of bone injury caused by uranium exposure is also poorly known. Thus, in the present study, the relevance of GJIC and the cytotoxic effect of MSCs induced by uranium were investigated. Cellular structural alterations were measured via transmission electron microscopy (TEM). Because membrane disruption was observed after uranium treatment, GJIC, a vital factor in cell signaling transduction, was further assessed to analyze whether GJIC was involved in the response of MSCs to uranium poisoning effects.

## 2. Materials and Methods

### 2.1. Cell Culture and Uranium Treatment

The human bone marrow derivative mesenchymal stem cell (hMSC) line was a gift of Dr. Yang (State Key Laboratory of Nuclear Physics and Technology, School of Physics, Peking University, Beijing, China). Cells were cultured in DMEM medium supplied with 7% fetal bovine serum, 15 ng/mL rhIGF-1, 125 pg/mL rhFGF-b, 2.4 mM L-Alanyl-L-Glutamine (Lonza, Walkersville, MD, USA).

For uranium treatment, the prepared 4.2 mM uranyl nitrate solution (UO_2_(NO_3_)_2_·6H_2_O), which is equivalent to 4.2 mM UO_2_^2+^, was stored as the stock solution (provided by the Research Institute of Nuclear Physics and Chemistry, China Academy of Engineering Physics, Mianyang, China). Working solutions were diluted to several concentrations (21 μM, 42 μM, 84 μM, 168 μM, 600 μM, 840 μM, and 1.4 mM) in culture medium before treatment. The concentration of uranyl nitrate used in the present work referred to the concentration used in the study of Jin et al., in which obvious γH2AX foci were produced when the concentration was beyond 10 μM [20]. Then, cells were incubated with medium containing uranyl nitrate for 24 h. Before assays, cells were washed twice with PBS and harvested. The control group underwent the same processes except they were exposed to uranium.

For uranium exposure experiments with the GJIC inhibitor, cells were cultured with 17 μM lindane (Sigma-Aldrich, St. Louis, MO, USA) which cannot impact cell viability in medium for 2 h (as shown in Appendix A), and then replaced with fresh medium supplemented with uranyl nitrate.

For uranium exposure experiments with different cell confluences, different quantities of cells were seeded in 96-well plates. A quantity of 10^4^ cells was seeded in each well and cultured for 24 h to reach 80% cell confluence with close connection. A quantity of 2 × 10^3^ cells was seeded in each well and cultured for 24 h to reach 10% cell confluence with sparse connection.

### 2.2. Cell Viability Assay

Cell viability was analyzed using a cell counting kit-8 (CCK-8) (Beyotime, Hangzhou, China). In brief, cells were cultured in 96-well plates at a density of 10,000 cells per well and treated with 21 μM, 42 μM, 84 μM, and 168 μM uranyl nitrate for 24 h. Then the medium with uranyl nitrate was discharged, and cells were washed three times with PBS and cultured with 1:10 CCK-8 solution in medium to detect the cell viability. For the analysis of uranyl nitrate inhabitation of cell proliferation, the treated cells were still cultured for 24 h post-exposure. Then, cell proliferation was assessed via CCK-8 assay. The optical density of each well was detected by a microplate reader (Spark, Tecan, Männedorf, Switzerland) at a wavelength of 450 nm. Data were obtained from at least three independent experiments.

### 2.3. Flow Cytometry Analysis

Annexin V-FITC/PI double staining was used to analyze the apoptotic cells after uranyl nitrate treatment. A quantity of 3 × 10^5^ cells was reseeded in 6-well plates and incubated with different concentrations of uranyl nitrate for 24 h; then cells were harvested and washed twice with ice-cold PBS (Beyotime, Hangzhou, China). Then, cells were suspended in 500 μL staining buffer containing 5 μL Annexin V-FITC and 10 μL PI (Annexin V-FITC/PI assay kit, Beijing 4A Biotech, Beijing, China). After being cultured for 10 min at room temperature in the dark, more than 10,000 cells were analyzed via flow cytometry (CytoFLEX, Beckman, Brea, CA, USA). The fraction of apoptotic cells was evaluated by FlowJo7.6.1 software.

### 2.4. Caspase3/7 Staining

CellEventTM Caspase-3/7 Green detection reagent (invitrogen, Eugene, OR, USA) was also used to indicate apoptosis. Cells were cultured in 96-well plates at a density of 10,000 cells per well and treated with 21 μM, 42 μM, 84 μM, or 168 μM uranyl nitrate for 24 h. Then cells were incubated with staining solution (4 μM) at 37 °C for 30 min. When the staining process finished, the plates were put in a microplate reader (Spark, Tecan, Männedorf, Switzerland) and the fluorescence was measured at 502 nm/530 nm.

### 2.5. Immunofluorescence Staining of γH2AX Foci for DNA Damage Assay

In present work, DNA double-strand breaks were indicated by γH2AX foci. The staining method was performed as previously described [32]. A quantity of 3 × 10^5^ cells was reseeded in a Φ25 mm glass-bottom dish (NEST, Woodbridge, NJ, USA) and treated with different concentrations of uranyl nitrate for 24 h. After treatment, cells were washed with phosphate buffered solution (PBS) (Beyotime, Hnagzhou, China), fixed with 4% formaldehyde (Sigma-Aldrich, St. Louis, MO, USA) in PBS for 20 min, and permeabilized with 0.5% Triton X-100/PBS for 20 min at 4 °C. After blocking with 1% bovine serum albumin (BSA) (Beyotime, China)/PBS, cells were incubated with γH2AX primary antibody (Abcam, Eugene, OR, USA) overnight and then labeled with Cy3-conjugated secondary antibody (Beyotime, Hangzhou, China) for three hours. After that, the samples were washed with PBS twice and stained with DAPI (Sigma-Aldrich, St. Louis, MO, USA). Fluorescence images were captured using an Olympus IX-83 (Olympus, Tokyo, Japan). More than 100 cells were scored by manual counting.

### 2.6. TEM Analysis

A quantity of 5 × 10^5^ cells cultured in a Φ60 mm dish was treated with different concentrations of uranyl nitrate for 24 h and harvested by a cell scraper at 24 h post-treatment. Cells were fixed with 0.5% glutaraldehyde (Sigma-Aldrich, St. Louis, MO, USA) at 4 °C for 10 min and postfixed with 1% OsO_4_ (Leica, Wetzlar, Germany) at 4 °C for 1.5 h. Then, cells were dehydrated in gradient ethanol solution and infiltrated with Epon812 (Beijing Zhongke Jingyi Technology Co., Ltd., Beijing, China) three times, once for 60 min, followed by embedding and curing. The embedded sample was sliced into a 50 nm film and double stained with uranyl acetate (Beijing Zhongke Jingyi Technology Co., Ltd., Beijing, China) for 15 min and lead citrate (Beijing Zhongke Jingyi Technology Co., Ltd., Beijing, China) for 2 min. Finally, the samples were observed using TEM (JEM-1400PLUS, JEOL, Tokyo, Japan).

### 2.7. Wound Healing Assay

To analyze the influence of uranyl nitrate treatment on the migration capacity of stem cells, cells were seeded onto the 6-well plates (3 × 10^5^ cells/well) and grown until reaching approximately 70% cell confluence. Cell monolayers were scratched with a sterile 200 μL pipette tip to make a wound. Then, cells were washed with PBS to remove cell debris and cultured for 24 h in medium supplied with 0 μM, 21 μM, 42 μM, 84 μM, or 168 μM uranyl nitrate solution. Cells were monitored under an Eclipse TS100 microscope (Nikon, Tokyo, Japan) and photographed at 0 h and 24 h. The efficient healing ratio was concluded from the width measured at 24 h divided by the result measured at 0 h. Each experiment was performed in triplicate. Data are showed as mean ± SD.

### 2.8. Transwell Migration Assay

Cells were treated with 0 μM, 21 μM, 42 μM, 84 μM, or 168 μM uranyl nitrate for 24 h. In each group, cells were harvested and rinsed three times. Approximately 1 × 10^5^ treated cells were reseeded onto transwell inserts (polycarbonate membrane with 8 μm pore size, Corning, Kennebunk, ME, USA) in 24-well plates with the culture medium. After 24 h, media within the transwell inserts were carefully removed. Cells inside the transwell inserts were removed by gently wiping with a cotton swab. Then, cells outside the transwell inserts were fixed with 5% paraformaldehyde and stained with crystal violet (Sigma-Aldrich, St. Louis, MO, USA). Migrated cells were viewed with a phase-contrast microscope (Nikon TS100, DS-U3 CCD, Tokyo, Japan). To quantitatively analyze the migrated cells, membranes containing the migrated cells were immersed in a 33% acetic acid solution to dissolve the crystal violet. The intensity of violet staining was determined as absorbance at 560 nm and measured via a microplate reader (Spark, Tecan, Männedorf, Switzerland).

### 2.9. GJIC Assay

Based on the established preloading/dye coupling method [33], the influence of uranyl nitrate treatment on the function of GJIC was evaluated. Briefly, 2 × 10^5^ donor cells cultured in a 12-well plate were treated with different concentrations of uranyl nitrate solution and stained with 5 μM calcein-AM (US EvervBright, Santa Clara, CA, USA) and 10 μM DiI (Beyotime, Hangzhou, China), and then rinsed with PBS three times for 5 min per time, trypsinized, and seeded onto 80% confluent recipient cells at a ratio of 1:200. After being maintained in incubators for 3.5 h, they were rinsed with culture medium to remove unattached cells and analyzed by a fluorescence microscope (IX83, OLYMPUS, Tokyo, Japan). GJIC levels were evaluated by the number of calcein-AM-positive recipient cells surrounded by donor cells. The number of calcein-AM-positive recipient cells was assessed by the intensity of fluorescence via a microplate reader (Spark, Tecan, Männedorf, Switzerland). The excitation and emission wavelengths for calcein-AM dye were 480 nm and 520 nm.

### 2.10. Western Blot Analysis

The experiment was carried out based on a standard protocol. Briefly, during the sample preparation, cells were lysed with ice-cold RIPA lysis buffer (Boster, Wuhan, China) with protease inhibitor (Thermo, Eugene, OR, USA) and phosphatase inhibitor (Boster, Wuhan, China). Protein concentration was measured by a bicinchoninic acid (BCA) protein assay kit (Boster, Wuhan, China); then cell lysate was maintained in a boiled water bath for 5 min. A quantity of 20 μg of total cellular protein was loaded onto SDS-polyacrylamide gel and the separated gel was transferred to 0.22 μm PVDF membranes (Millipore, Billerica, MA, USA), and then blocked in 5% BSA and blotted with polyclonal connexin43 antibody (Abcam, Eugene, OR, USA), polyclonal connexin32 antibody (Abcam, Eugene, OR, USA) and polyclonal GAPDH antibody (Boster, Wuhan, China)) (1:1000 diluted in 5% BSA). Membranes were incubated at 4 °C overnight, washed three times with TBST (0.1% Tween) for 15 min, and then hybridized with horseradish peroxidase-conjugated secondary antibodies (1:5000 diluted in 5% BSA) for 1 h at room temperature. Then, membranes were washed three times with TBST (0.1% Tween) for 15 min. The protein bands were visualized with chemiluminescent reagent (Boster, Wuhan, China) and their images were obtained through a luminescent image analyzer (ChemiScope 6000Exp, CLiNX, Shanghai, China).

### 2.11. Statistical Analysis

Data from at least three independent experiments are presented as mean ± SD. Statistical comparisons among different groups were carried out via one-way ANOVA, followed by the Student–Newman–Keuls test.

## 3. Results

### 3.1. Detrimental Effects Observed in hMSCs after Exposure to Uranyl Nitrate

TEM micrographs of cells treated with different concentration uranyl nitrate were obtained and are shown in Figure 1A. In the control group, subcellular structures, such as cellular nuclei, membranes, and mitochondria, are clearly represented in the TEM micrographs. Obvious exosomes and apoptotic bodies appeared at 84 μM and the outline of the cellular membrane became obscure compared to that of the 42 μM treatment group. In addition, with the increasing concentration of uranyl nitrate, evident elevation of lysosome number and serious disruption of the cellular membrane were observed at 420 μM.

After hMSCs were exposed to uranyl nitrate for 24 h, cell viability showed a dose-dependent relationship (Figure 1B). In the comparison with the control group, a slight decrease in cell viability was detected at uranyl nitrate concentrations below 84 μM (0.99 for 21 μM, 0.96 for 42 μM). With the increasing concentration, cell proliferation had a steeper descent. The results of apoptotic cell proportion showed that a significant enhancement was detected at a concentration of 84 μM (1.13-fold in comparison with the control group, *p* < 0.05) (Figure 1C), in which a remarkable disruption of the cellular membrane was produced, as shown in TEM micrographs (Figure 1A).

As cell viability and apoptosis are subsequent events regulated by DNA damage response, DSBs, a crucial indicator of DNA damage, were further measured via γH2AX staining at 30 min and 24 h post-uranyl nitrate treatment (Figure 1D). An increasing yield of γH2AX foci was detected at 30 min with the enhancement of concentration. At 24 h post-uranyl nitrate exposure, γH2AX foci number decreased, but was still remarkably higher than that of the control group in all uranyl nitrate-treated groups, except at the lowest concentration of 21 μM.

### 3.2. The Relevance of GJIC to Cytotoxic Effects of hMSCs Induced by Uranyl Nitrate

Considering the disruption of the cellular membrane can influence the function of GJIC, apoptosis caused by uranyl nitrate in high or sparse cell confluence was analyzed. Results of flow cytometry analysis showed that the proportion of Annexin V-FITC^+^PI^+^ increased with uranyl nitrate concentration. The high proportion of Annexin V-FITC^+^PI^+^ population was 5.37% for 1.4 mM uranyl nitrate with high cell confluence or 18.9% for 1.4 mM uranyl nitrate with sparse cell confluence. This indicates an enhanced death percentage was observed with increasing concentration of uranyl nitrate no matter the cell culture confluence. However, at the same concentration of uranyl nitrate, the death proportion in sparse growth was much higher than that in high confluent growth (Figure 2A). A similar trend was detected in cellular viability assay (Figure 2B), when cells were treated with a low concentration of uranyl nitrate at high and sparse cell confluence. In the comparison with the control group, cell viability was 0.85 for 84 μM at high confluence and 0.71 for 84 μM at sparse confluence. Moreover, the higher cell viability after uranyl nitrate exposure observed in the high confluent growth group disappeared when cells were pretreated with lindane, a GJIC inhibitor (Figure 2C). This showed that pretreatment with lindane reduced cell viability to 0.52 for 84 μM at high confluence in comparison with the lindane-pretreated control group.

Further, scratch wound healing assay was carried out to assess the influence of uranyl nitrate exposure on the migration ability of hMSCs. The representative pictures showed that the efficiency of wound healing of hMSCs was impacted after uranyl nitrate exposure (Figure 3A). Comparing the width of scratches between 0 h and 24 h, the healing rate can be quantified and the difference induced by different concentrations of uranyl nitrate can also be statistically analyzed. The quantification of the healing rate revealed that after uranyl nitrate treatment, the wound healing rate was reduced in comparison to that of the control group (56.70% for 21 μM vs. 73.39% for the control group). The impairment of wound healing exhibited a dose-dependent increasing behavior with uranyl nitrate concentration (Figure 3C). To confirm the impairment in migration ability, transwell migration assay was also performed as a parallel experiment. The uranyl nitrate-treated cells were reseeded onto transwell inserts (polycarbonate membrane with 8 μm pore size, Corning) in 24-well plates and cultured for 24 h. Cells outside of transwell inserts were fixed with 5% paraformaldehyde and stained with crystal violet (the representative pictures shown in Figure 3B). In order to quantify cells outside of transwell inserts, cells stained with crystal violet were dissolved in 33% acetic acid solution and the OD_560nm_ value was measured by a microplate reader. The quantification results of the OD_560nm_ value showed that the quantity of migrated hMSCs declined after uranyl nitrate treatment, and statistically significantly declined at 168 μM (OD_560nm_ = 0.05 for 168μM vs. OD_560nm_ = 0.08 for control group, *p* < 0.05) (Figure 3D).

The expression of connexin43 and connexin32, proteins related to GJIC function, was also measured to analyze the influence of uranyl nitrate exposure on GJIC function. It showed that at 30 min post-uranyl nitrate treatment, the expression of connexin43 and connexin32 decreased with the increasing concentration, which was more obvious at a high concentration of 84 μM. The results at 24 h post-uranyl nitrate treatment showed an increasing expression of connexin43 compared with that at 30 min, but this was still lower than that of the control group at 84 μM. The expression of connexin32 at 24 h had a similar trend. Furthermore, GJIC function was also analyzed via the dye coupling method. Cells treated with uranyl nitrate of different concentrations were stained with dye passage and Dil. Then, they were seeded as donor cells in uranyl nitrate-untreated recipient cells. The recipient cells coupled with green fluorescence that were transferred from donor cells were counted. The results showed that exposure to uranyl nitrate for 24 h significantly decreased the number of recipient cells with green fluorescence (Figure 4B,C).

## 4. Discussion

After intake, uranyl ions can be absorbed by the kidneys within 24 h and the residual is mainly deposited in the bones (10–15%) [34], which can induce bone dysfunction to threaten human healthy. Increasing studies indicate MSCs achieve therapeutic effects in tissue regeneration, including bone injury [35]. Clarifying the damage mechanism of hMSCs after uranium exposure can be beneficial in exploring new strategies for curing uranium poisoning. Therefore, the cytotoxic effects of hMSCs after uranium exposure were examined in the present work.

TEM micrographs showed that subcellular organelle disruption occurred with the increase in uranyl nitrate concentration (Figure 1A). Moreover, an evident decreasing cell viability (Figure 1B) and apoptotic effect (Figure 1C) appeared at 84 μM, in which an obvious disruption of cell membranes, apoptotic bodies, and abundant exosomes were observed in TEM micrographs (Figure 1A). Furthermore, DSBs, a crucial indicator of DNA damage, were detected via γH2AX staining with immunofluorescence assay. The results showed that, similar to the results of cell viability, the production of γH2AX foci caused by uranyl nitrate exposure was dose dependent (Figure 1D). At 24 h post-treatment, a high quantity of residual foci was detected, showing a slight DNA damage repair process. This was totally differentiated from the resolution of γH2AX foci induced by γ-ray radiation [20]. This indicated that subcellular organelles, as well as cell nuclei, were impaired by uranium exposure. It has been demonstrated that uranium can localize inside lysosomes in which uranium precipitates with phosphate and forms microcrystals at high concentrations [36,37]. Lysosome not only participates in apoptosis regulation, but also mediates cell death through autophagy [38]. Pierrefitecarle et al. reported that exposure to uranium for 3 h rapidly activated autophagy in osteoblastic cells but depressed autophagic flux to accumulate autophagic vesicles for 24 h [39]. This shows that, although the cell nucleus is the most important target for ionizing radiation, subcellular organelles are alternative targets under uranium exposure.

Considering that the completeness of cellular membranes is a basis for GJIC to participate in cell signaling transduction, there is the potential for GJIC to influence uranium poisoning effects, because an obvious disruption of cellular membranes, accompanied with significant apoptotic effects, was observed in our work, as mentioned above. GJIC function is depressed in sparse cell confluence because few connections exist between cells. Therefore, apoptosis of cells in high and sparse cell confluence was measured to analyze whether GJIC is associated with cytotoxic effects induced by uranium. The apoptotic proportion indicated by the Annexin V-FITC^+^PI^+^ population increased with the elevation of uranyl nitrate concentration. Interestingly, the apoptosis proportion in the sparse cell confluence group was remarkably higher than that of the close connection group (Figure 2A). This detrimental effect was confirmed by the impairment of cell viability, as the results showed there was lower cell viability in sparse cell confluence than in the closely contacted group (Figure 2B). Importantly, when hMSCs were pretreated with a GJIC function inhibitor, lindane, an obvious impairment of cell viability appeared in close connection, which was even lower than that of the sparse cell confluence group. This suggests that the inhibition of GJIC can strengthen the toxic effects of uranium in hMSCs.

Further, the damage of GJIC function after uranyl nitrate exposure was also assessed via the analysis of hMSC migration impairment, as hMSC proliferation and migration are maintained by GJIC function [40,41]. The migration ability was analyzed via wound healing and transwell migration assay. The results showed that the efficiency of wound healing was suppressed after uranyl nitrate exposure, and the quantity of migrated cells shown by transwell migration assay evidently declined at the concentrations of more than 84 μM (Figure 3). This suggests the disruption of cell membranes caused by uranyl nitrate has the potential to result in the impairment of GJIC. To verify this, the expression of gap functional proteins connexin43 and connexin32 was measured after hMSCs were exposed to uranyl nitrate of different concentrations (Figure 4A). A dose-dependent depression of connexin43 and connexin32 was observed after hMSCs were exposed to increasing concentrations of uranyl nitrate. Connexin43 was reported to be involved with plasma membrane internalization and degradation events, and opening and closing of the channel [42,43]. Because the functionality of connexin43 allows the exchange of the dye between donor and recipient cells, the influence of uranium exposure-induced suppression of connexin43 expression on GJIC function was assessed by a dye transfer experiment (Figure 4B,C). There was a reduced proportion of recipient cells with green fluorescence that transferred from uranyl nitrate-treated hMSCs preloaded with dye, suggesting GJIC function of hMSCs was influenced by uranyl nitrate. It has been reported that GJIC plays a role in propagating the DNA damage repair signal from bystander cells to irradiated cells, and in enhancing the DSB repair in irradiated cells [44]. Thus, the GJIC dysfunction induced by uranyl nitrate might contribute to the high residual γH2AX foci number observed at 24 h post-uranyl nitrate treatment. Due to uranyl nitrate owning chemical characteristics like those of heavy metal, it has the possibility of directly interacting with protein and impacting its biological function. For example, Yu et al., through density functional theory calculations, demonstrated that Th^4+^ could form a stable, octa-coordinate structure at the sites of respiratory chain complex IV, which led to ultimate cytotoxic effects [45]. Hussain et al. reported that arsenic can directly incorporate with connexin43 and result in its structural conformational changes [46]. Thus, unlike in low-dose ionizing radiation, GJIC, being a sensor transducing cellular signal, can be a target that can be directly impacted by uranyl nitrate in uranium exposure. In the present work, the results showed there was a relevant GJIC function and cytotoxic effects induced by uranyl nitrate. However, the molecular mechanism involved in GJIC mediating uranyl nitrate poisoning effects is complex, and over the past two decades it has been demonstrated that connexins can also act as ‘hemichannels’ enabling communication between cytosolic compartments and the extracellular environment [47]. Although these connexin hemichannels are usually maintained in a closed state under physiological conditions, mechanical stimulation, inflammation, and other pathological conditions can stimulate their opening [48,49,50]. Therefore, more studies are needed to identify the subsequent events triggered by GJIC dysfunction induced by uranium in order to illuminate the mediating mechanism.

## 5. Conclusions

In summary, the cytotoxic effects of hMSCs induced by uranyl nitrate were evaluated in the present study. The results showed that an obvious elevation in the apoptotic cell population was caused by a concentration in which a significant disruption of cell membranes was obtained in a TEM micrograph. A higher apoptosis of hMSCs induced by uranium exposure was obtained in sparse cell confluence than in closely contacted cell confluence. Inhibition of GJIC caused by lindane pretreatment efficiently impaired cell viability in the closely contacted group. In accordance with this, after uranium exposure, the expression of GJIC functional proteins connetxin43 and connexin32 was suppressed, which was accompanied by a depression in the cell migration ability. This implies GJIC function was damaged during uranyl nitrate exposure. These findings suggest the GJIC function can be a target of uranyl nitrate exposure and influence the poisoning effects of uranium.

## Figures and Tables

**Figure 1 biology-13-00525-f001:**
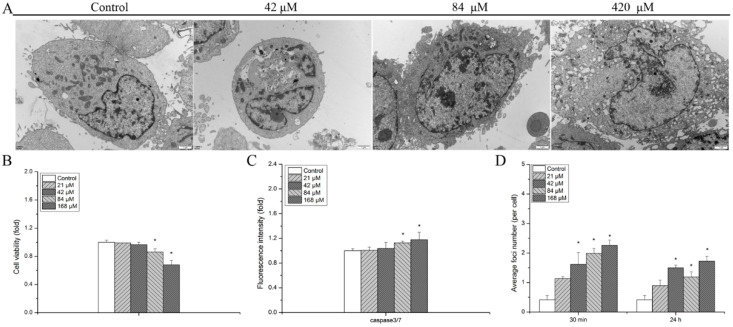
The detrimental effects induced by uranyl nitrate. (**A**) TEM micrograph of hMSCs after being exposed to uranyl nitrate with different concentrations for 24 h; (**B**) the cell viability was measured via CCK-8 assay at 24 h after hMSCs were cultured with different concentrations of uranyl nitrate for 24 h, and data were normalized to the control group; (**C**) cell apoptosis was measured at 24 h after cultured with different concentrations of uranyl nitrate for 24 h, and the results were normalized to the control group; (**D**), quantification analysis of γH2AX foci induced by uranyl nitrate with different concentrations at 30 min and 24 h post-treatment. The representative picture of cells with γH2AX foci produced by different concentrations of uranyl nitrate are shown in Appendix A. All data are presented as the mean ± SD (* represents *p* < 0.05).

**Figure 2 biology-13-00525-f002:**
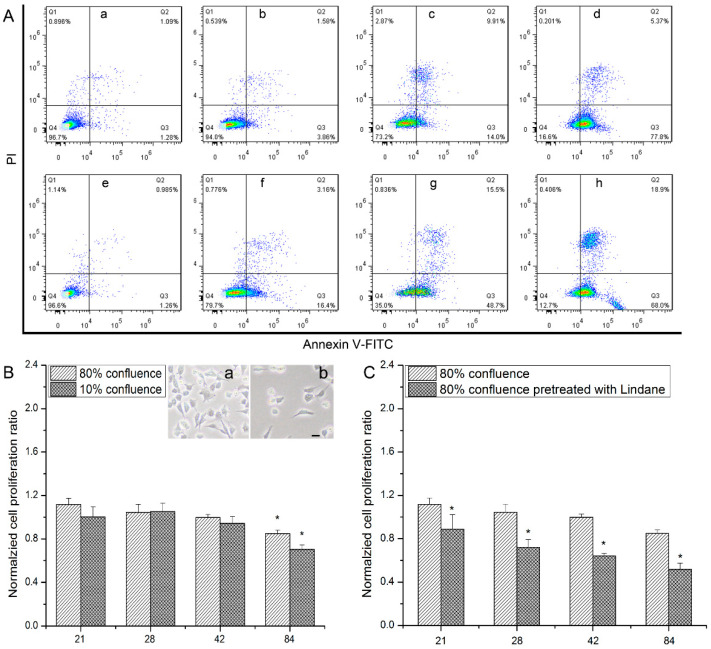
The influence of cell confluence on the damage effects induced by uranyl nitrate. (**A**) Flow cytometry analysis of apoptosis of hMSCs that were cultured with different concentrations of uranyl nitrate for 24 h. (**a**–**d**) Cells were cultured with 0 μM, 600 μM, 840 μM, and 1.4 mM uranyl nitrate at high (80%) contact cell confluence. (**e**–**h**) Cells were cultured with 0 μM, 600 μM, 840 μM, and 1.4 mM uranyl nitrate at sparse (10%) contact cell confluence. (**B**) The results of cellular proliferation induced by different concentrations of uranyl nitrate at sparse (10%) and high (80%) cell confluence. The upper-right panel in Figure 2B shows the representative pictures of cells at high (80%) (**a**) and sparse (10%) (**b**) cell confluence. Bar = 15 μm. Cellular proliferation was measured via CCK-8 assay at 24 h after exposure to different concentrations and OD value was normalized to the control group. Data were normalized to the control group. (**C**) The influence of GJIC inhibition on cellular proliferation after uranyl nitrate treatment at high (80%) cell confluence. hMSCs were treated with 17 μM lindane for 2 h before uranyl nitrate treatment to inhibit GJIC. Cellular proliferation was detected at 24 h post-uranyl nitrate treatment. Data of uranyl nitrate treatment groups were normalized to the control group. Data of the lindane pretreatment group were normalized to the control group pretreated with lindane. All data are presented as the mean ± SD (* represents *p* < 0.05).

**Figure 3 biology-13-00525-f003:**
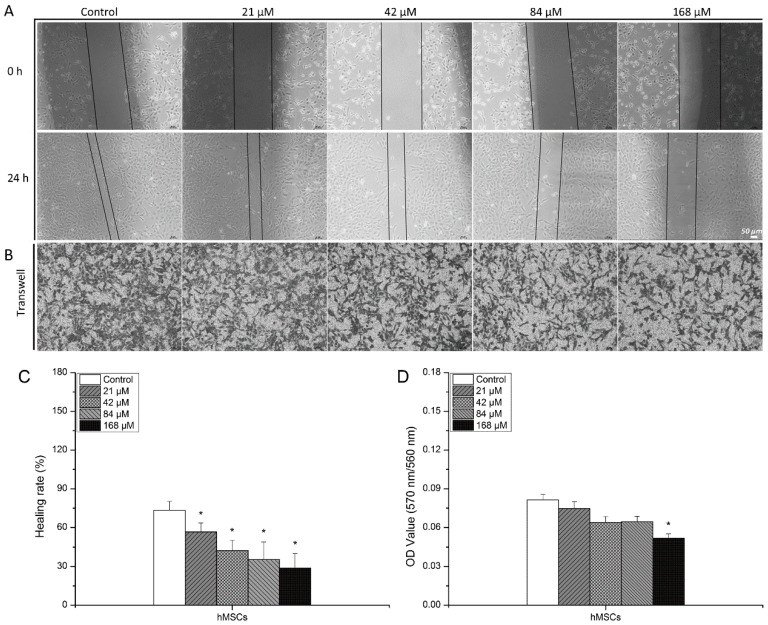
The influence of uranyl nitrate treatment on the migration ability of hMSCs. (**A**) Scratch wound healing assay was used to assess the effect of uranyl nitrate treatments on stem cell migration. Cell confluence was wounded by a pipette tip and washed with medium twice. Then cells were cultured with uranyl nitrate of different concentrations for 24 h, except for the control group. Cells were photographed immediately (0 h) after a wound appeared and 24 h later. (**B**) The representative picture of migrated hMSCs after hMSCs were exposed to uranyl nitrate. Transwell migration assay was used to analyze the effect of uranyl nitrate treatment on stem cells’ invasion ability. After hMSCs were exposed to uranyl nitrate for 24 h, cells were harvested and 1.0 × 10^5^ cells/well were seeded onto 8 μm transwell inserts in 24-well plates and continuously cultured for 24 h. Migration was assessed by staining cells with crystal violet, which was viewed under a phase-contrast microscope. Bar = 50 μm. (**C**) Quantitation analysis of scratch wound healing efficiency impacted by uranyl nitrate exposure. In each picture of Figure 3A, the leading edges of the scratch are depicted by the black straight line. The width between each leading edge of the scratch was measured via Image J 1.8.0 software and the efficient healing ratio was concluded from the width measured at 24 h divided by the result measured at 0 h. (**D**) Quantitative analysis of the quantity of migrated cells. Membranes containing migrated cells were cut out and immersed in a 33% acetic acid solution to dissolve the crystal violet. The intensity of violet staining was determined as absorbance at 560 nm. All data are presented as the mean ± SD (* represents *p* < 0.05).

**Figure 4 biology-13-00525-f004:**
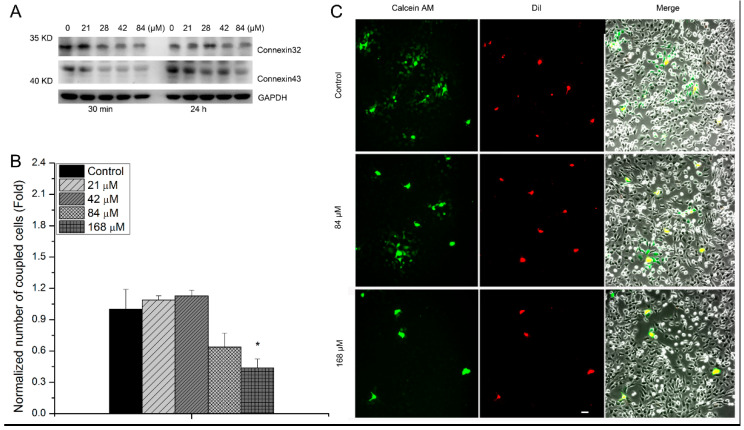
The suppression of GJIC function in hMSCs after exposure to uranyl nitrate. (**A**) The expression of connexin43 and connexin32 proteins via Western blot analysis after hMSCs with close cell contact were treated with uranyl nitrate of different concentrations. The cells were harvested at 30 min or 24 h post-uranyl nitrate treatment. The original imaging of protein bands for connexin32 and connexin43 were supplied in Appendix A. (**B**) Quantification of the number of coupled dye cells per donor cell. The dye (calcein-AM, green) was transferred from donor cells to unlabeled recipient cells and the intensity measured by a microplate reader with Ex/Em = 480 nm/520 nm. The red fluorescence represents hMSCs cultured with uranyl nitrate for 24 h. (**C**) Representative pictures showing the concentration-dependent influence of uranyl nitrate on the GJIC function through the dye coupling assay. Bar = 50 μm. All data are presented as the mean ± SD (* represents *p* < 0.05).

## Data Availability

All data generated or analyzed during this study are included in this published article. Requests for data and materials should be addressed to Yi Quan (yiquan_206_inpc@163.com).

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
