# Peer review of "The Cytotoxic Effects of Human Mesenchymal Stem Cells Induced by Uranium"

_biology, 2024, doi:10.3390/biology13070525_

Round 1

Reviewer 1 Report

Comments and Suggestions for Authors

The manuscript of Yi Quan et al. addressed an interesting topic regarding the toxic effects of uranium on mesenchymal stem cells (MSCs). Using an in vitro model, involving human mesenchymal stem cells (hMSCs) exposed to uranium (i.e., uranyl nitrate), the authors observed that the susceptibility of these cells to uranium was influenced by gap junction intercellular communication (GJIC), particularly connexin43-related GJIC communication.

While the findings of the study are novel, the experimental approach of the study and certain aspects of the research are not fully clear to the reviewer. Therefore, further revision and improvement of the manuscript are needed in order to meet the quality standards of Biology.

·       It would be beneficial for the manuscript to incorporate some information related to GJIC and connexins in the introduction section.

·       More details about the cell line selected for the study, as well as references in which this cell line has been used, should be provided.

·       Further information about the dose of uranyl nitrate selected for the study should be included in the manuscript. Is this dose biologically relevant? No supporting references are cited in the manuscript.

·       The manuscript should specify the solvent in which uranyl nitrate was dissolved, as well as the final concentration of the solvent used in the assays.

·       The authors should standardize the concentrations of uranyl nitrate evaluated in the assays. In the materials and methods section, several concentrations are mentioned (e.g. line 86). However, these concentrations differ in each section and figure of the manuscript (e.g. Figure 1A up to 420 uM, Figure 1B up to 50 uM, Figure 1C up to 84 uM, Figure 1D up to 168 uM, Figure 2D 600, 840 and 1400 uM).

·       The authors should justify why 17 uM of lindane was selected as a GJIC. No supporting references are cited in the manuscript.

·       It is unclear what cell density was used in each of the described assays. The authors stated in line 94 that they employed 2 different cell seeding densities to achieve varying degrees of confluency. However, when these densities were applied is not clear to the reviewer. A probe of this is that in the section on cell viability and alternative cell seeding density is mentioned. This also applies to the flow cytometry analysis section. On the other hand, no cell seeding densities are stated in the caspase3/7 or immunofluorescence staining sections. Additionally, it would be beneficial for the manuscript to include a representative image of hMSC cultures at low and high confluency as supplementary material.

·       The authors should maintain consistency in the level of detail provided for each assay described in the materials and methods section (e.g. in some assays, it is specified how the cells were harvested, whereas in others, this information is less clear). Moreover, important details regarding western blot analysis are missing (e.g. how the membranes were blocked, how the membranes were washed, for how long the primary and secondary antibodies were incubated, and at what temperature).

·       The title of the section 2.5. should be changed to a more clear title (e.g. DNA integrity analysis).

·       More information about the products and reagents used in the research should be provided, including specific examples like calcein AM, Annexin V/PI, or DiL.

·       The identification code and dilution of the anti-connexin 43 and anti-GAPDH antibodies should be indicated in the Materials and Methods section. Additionally, information about the secondary antibody and its dilution should be included in the manuscript.

·       Graphical information, such as representative images, of the yH2AX loci staining should be included in the manuscript to support the quantification indicated in Figure 1D.

·       The term “intensity confluence” should be rephrased as “high confluence”.

·       The effect of the GJIC inhibitor lindane is not clear to the reviewer. The authors used it as a positive control in section 3.2, but its actual effect in a scrape loading assay (Figure 4) is missing. To provide concrete results, the authors should include the effect of lindane in the scrape loading assay, otherwise the results observed in Figure 2 are merely speculative.

·       Figures 2 B and C should include the cellular proliferation ratio at a 0 uM concentration of uranyl nitrate. This is especially important for Figure 2C to clarify if the blockage of the GJIC by lindane can also impair cell proliferation in the absence of uranyl nitrate.

·       It is not clear to the reviewer at what cell confluency the experiments for quantifying connexin43 expression were performed. This information is crucial since cell confluency can affect connexin43 expression.

·       The full western blot shown in Figure 4 should incorporated into the manuscript. Additionally, it would be informative if the authors could measure the various posttranslational modifications observed in the connexin43 western blot. Connected to this, the authors might include an immunofluorescent staining of connexin43 to clarify its subcellular localization. It is widely recognized that this protein can also be present in particular cellular organelles (e.g. mitochondria) as well as in the cytosol of cells.

·       Despite the novelty of the results, the conclusions drawn by the authors seem to be somewhat speculative. How can the impairment of GJIC affect the cytotoxic effects of uranyl nitrate? Do these connexin43 gap junctions serve as a means for the cells to counteract the cytotoxic effects? If so, is there evidence in other similar pathologies of how this can be achieved? What molecules can be transferred via GJIC that can counteract the cytotoxic effects?

·       Furthermore, the authors ought to explain why they have focused their research on connexin43 and not on other connexin species (e.g. 32, 26...) that can also mediate GJIC. Is there evidence in the literature that hMSCs do not express any other connexin isoforms?

·       The authors focused primarily on connexin43-mediated GJIC. However, it is worth noting that connexins can also play a role in intercellular communication through the formation of connexin hemichannels. This alternative form of cellular communication is typically observed under pathological conditions, and therefore, it would be beneficial for the manuscript to include information about this in the discussion.

Comments on the Quality of English Language

·       The manuscript contains several typos and inconsistencies, including incorrect abbreviations and scientific notation (e.g. cell densities). It is recommended that the authors carefully review and revise the manuscript to address these issues.

·       The use of non-scientific terminology throughout the manuscript (e.g. 'tiny', 'attack', 'and so on') should be replaced with formal scientific language.

·       Several sentences in the introduction and discussion are difficult to understand and should be rephrased to improve readability.

Reviewer 2 Report

Comments and Suggestions for Authors

The cytotoxic effects of human mesenchymal stem cells induced by uranium

Yi Quan et al; In this manuscript, the toxicity of uranyl nitrate was elucidated using human mesenchymal stem cells. They designed experiments well. In this article, they found molecular targets for uranium toxicity and assessed cell morphology using TEM analysis, cell viability, apoptosis, and DNA damage events. Comments can be found below.

 1. Authors can improve the title of the article.

2. A gentle decrease in cell proliferation was examined at concentrations of 21 μM and 28 μM. Authors should indicate how severe the decline in prevalence is, rather than gentle.

3. The results of apoptotic cell proportion showed that compared to the control group, a slight but notable increase was observed at a concentration of 84 μM (Figure 1C), at which there was significant disruption of the cell membrane, as shown in TEM images. The author should indicate the value and avoid using frivolous and gentle expressions in the results section.

4. Authors should provide the images of γH2AX foci.

5. Fig. 1 B, viability assay, data on higher concentrations of uranyl acetate are missing, authors should report viability at higher concentrations.

6. The results of flow cytometry analysis showed that the proportion of annexin V-FITC+PI+ increased with uranyl nitrate. Authors must report the proportion of apoptotic numbers.

7. A similar trend was observed in the cell viability assay (Figure 2B). Authors should elaborate on the results in more detail.

8. Pretreatment with GJIC inhibitor resulted in the cell viability of hMSCs after uranyl nitrate exposure being lower than at confluence of thin cell cultures. Authors should indicate the change in viability.

9. Figures 3C and 3D, requires further explanation in the results section

10. Figure 3A: Authors should provide time points of 6 and 12 hours to support the results.

11. Fig. 4A. Authors should quantify intensity.

12. The results showed that there was an inhibition of connexin43 expression after treatment with uranyl nitrate, which was increased with the increased concentration. After recovery, connexin43 expression was still suppressed at 24 h, except for the lowest concentration of 21 μM (Figure 4A). Authors should rewrite and elaborate the sentence.

13. Authors can measure the membrane potential of uranyl-treated cells to amplify proposed molecular events related to toxicity.

Reviewer 3 Report

Comments and Suggestions for Authors

In here, the authors suggest that possible toxicity to mesenchymal stem cell in bone by Uranium. Although the limited amount of uranium have been established, the possible toxic effects are needed to study. The study was designed well and it is in scope of this journal. 

Author Response

Dear Editor and Reviewers,

First and foremost, we would like to express our sincere gratitude for the valuable comments and suggestions provided on our study. We are delighted to hear that our research design has been acknowledged as appropriate and that it falls within the scope of your esteemed journal.

Regarding the concerns raised about the potential toxicity of uranium to mesenchymal stem cells in bone, we fully concur that further investigation into the possible toxic effects is warranted, despite the established limited quantities of uranium.

We recognize that while our current study has explored the impact of uranium on mesenchymal stem cells, there is indeed much more to uncover. We plan to expand our work in this area in future research endeavors to comprehensively understand the mechanisms of uranium toxicity and its potential health risks.

Once again, we appreciate the feedback. We look forward to continuing our collaboration with the journal and delving deeper into our research.

Yours sincerely,

Yi Quan

Research Associate

China Academy of Engineering Physics

Round 2

Reviewer 1 Report

Comments and Suggestions for Authors

Dear Editor and Authors,

I would like to thank the authors for their efforts in revising the manuscript and addressing the reviewer's comments. While the recent improvements to the manuscript are commendable, there are still several issues that need to be addressed before it can be published in Biology.

·      Comment 10: Representative images of the yH2AX loci staining for all the conditions evaluated should be included in the manuscript to support the quantification presented in Figure 1D. These images should be included as supplementary material rather than as a small picture in Figure 1. In addition, the microscopic images should be accompanied by a scale bar.

·      Comment 11: Although lindane is widely used as a GJIC blocker in the literature, the authors should provide evidence in their GJIC dye coupling assay that lindane significantly impacts the GJIC of their cells. Including lindane as a positive control in Figure 4 is necessary to confirm that the reduced number of coupled cells observed in the different doses of uranyl nitrate is directly linked to a suppression of the GJIC.

·      Comment 13: The full western blots of the Cx43 and Cx32 expression levels shown in Figure 4 should be included in the manuscript as supplementary material.

Comments on the Quality of English Language

·      Comment 18: The manuscript should undergo thorough revision to address several inconsistencies and typographical errors. For instance, in line 65, the authors indicated "become" twice, and in line 66, "cytotoxicology" is misspelled.

Reviewer 2 Report

Comments and Suggestions for Authors

The authors took into account most of my comments and made the most necessary corrections.. Although, not all of my questions have been explained entirely satisfyingly, among other the quality of image H2Ax could be improved. I appreciate the efforts made by the authors to improve the text In the reviewer's opinion in the present form, the work meets the requirements for articles published in the Biology and I recommended its publication
